# Access to HIV prevention, treatment, and care services during COVID-19 by men who have sex with men in Zimbabwe, An interpretive phenomenological analysis study

Idah Moyo[1,2]*, Azwihangwisi Hellen Mavhandu-Mudzusi[3], Freddy Lipeleke[4], Musa Sibindi[4,5]

1 Department of HIV Services, Populations Solutions for Health, Harare, Zimbabwe, 2 Department of Health Studies, University of South Africa, Pretoria, South Africa, 3 Office of Graduate Studies and Research, University of South Africa, Pretoria, South Africa, 4 Department of Sociology, University of Johannesburg, Johannesburg, South Africa, 5 Sexual Rights Centre, Bulawayo, Zimbabwe

* idahbandamoyo@gmail.com

**Data Availability Statement:** All relevant data are incorporated in the manuscript and its Supporting information files.

## Abstract

### Background

Key populations are disproportionately affected by the human immunodeficiency virus (HIV) but have less access to HIV prevention and treatment services. The Coronavirus disease-2019 (COVID-19) pandemic is reinforcing health inequities among vulnerable populations, including men who have sex with men (MSM). Therefore, this article presents the findings describing the experiences of MSM in accessing HIV services during COVID-19 in the second largest city of Zimbabwe.

### Methods

An interpretative phenomenological analysis design was applied to gain some insights regarding the lived experiences of MSM in accessing HIV prevention, treatment, and care services during COVID-19 lockdowns era in Zimbabwe. Data were collected from 14 criterion purposively selected MSM using in-depth, one-on-one interviews. Data were analysed thematically guided by the interpretative phenomenological analysis framework for data analysis.

### Results

The findings demonstrated that during the COVID-19 lockdowns in Zimbabwe, MSM faced several barriers as they tried to access HIV services. Some of the barriers included the need for travel authorisation letters and treatment interruption. The study also found that COVID-19 and related restrictive measures had psychosocial and economic effects that encompassed loss of income, intimate partner violence and psychological effects.

**Funding:** The authors received no funding for this work.

**Competing interests:** The authors have declared that no competing interests exist.

## Conclusions

Limited access to healthcare services by MSM due to COVID-19 lockdown may negatively affect the viral suppression and fuel the spread of HIV, which may reverse the gains toward the control of HIV epidemic. To sustain the gains toward HIV epidemic control and to ensure continuity of treatment, particularly for members of key populations, it is critical that the health-care delivery system adjusts by taking service to the community through adopting a differentiated service delivery approach.

## Introduction

In response to Coronavirus disease-2019 (COVID-19), countries were compelled to implement lockdown measures to contain the pandemic. These measures had a negative impact on access to HIV services [1]. The Joint United Nations Programme on HIV/AIDS (UNAIDS) [2] indicates that key populations are particularly vulnerable to COVID-19, HIV acquisition and HIV service interruptions. Examples of key populations include the following: men who have sex with men, sex workers, transgender people, people who inject drugs and people in prisons and other closed settings. UNAIDS [2] further defines men who have sex with men as referring to all men who engage in sexual and/or romantic relations with other men. A study conducted in Kenya found that HIV clinics for MSM during the March to August 2020 lockdown were either closed down or suffered some form of disruption as they were deemed as not providing essential services [3]. This was also the case in Zimbabwe [4]. Therefore, gaps in accessing HIV services may widen and the gains made in HIV programming for key populations are at risk of being reversed as a result of disruptions caused by COVID-19 [2].

In the given context it remains critical to ensure access to HIV prevention and treatment services for key populations during COVID-19.

Key populations are severely affected by the human immunodeficiency virus (HIV) compared to the general population [5]. Despite the effective implementation of the HIV prevention, treatment and care programme in Zimbabwe and the significant progress that the country has made, the key population groups have not been fully embraced [5]. Evidence indicates that men who have sex with men have unmet HIV prevention and treatment needs and as a result this group has accounted for new infection [5]. Statistics indicate that in 2020, key populations and their sexual partners accounted for 65% of HIV infections globally and 39% of infections were in sub-Saharan Africa [6].

With regards to MSM, the of risk of acquiring HIV was 25 times higher than that of all adult men. Furthermore, 23% of new infections were from this group globally [6]. To achieve HIV epidemic control, it is critical that key populations have access to HIV care services, which include testing, treatment and retention in care and maintenance of an undetectable viral load [7].

The emergence of COVID-19, together with the associated national lockdowns and restricted movement, has brought diverse challenges that further aggravate the situation for key populations. The criminalisation of same sex relationships (in Zimbabwe for example), generates stigma and discrimination, making it difficult to access HIV services [8]. To curb the spread of COVID-19, the World Health Organization (WHO) called for adjustments in HIV care service delivery [9]. These adjustments include among others, expediting differentiated HIV service delivery through the use of strategies such as telehealth, multi-month dispensing

of antiretroviral therapy (ART) and pre-exposure prophylaxis to prevent treatment interruptions [1, 10]. Despite these modifications, there is evidence that key populations experience challenges in navigating their way to treatment facilities [11, 12]. In addition, one study [11] indicated that under COVID-19 pandemic conditions, people living with HIV are likely to experience ART service interruptions, particularly where there are fragile health systems.

The 2025 UNAIDS targets population groups that are difficult to access like key populations in the prioritisation of HIV services [13]. COVID-19 has had a major impact and setback in rolling back HIV [14]. In the context of COVID-19 and related restrictive measures, it is estimated that a six-month interruption on the supply of ART would result in 1, 63 times deaths over a one-year period [15]. COVID-19 induced reversal of HIV response is demonstrated by a study that showed that HIV testing dropped by 20% globally [16], with a significant decrease in pre-exposure prophylaxis and a reduction in condom access [17]. There are also reports of reduced access to ART due to barriers associated with COVID-19 restrictive measures and fears of COVID-19. All these have had a negative impact on viral load suppression for people living with HIV [18].

Key population, like MSM faced barriers in accessing HIV care services during COVID-19, a situation aggravated by social and legal issues peculiar to them, thus increasing their vulnerability [19]. It has been argued that fundamental structural barriers such as stigma and discrimination against key population have been compounded by COVID-19 restrictions, putting them at risk and forcing them to defer access to HIV services [18].

Studies conducted in Zimbabwe and elsewhere before the start of the COVID-19 pandemic demonstrated that MSM were already experiencing barriers in accessing HIV care services [20–22]. The study was justified, given the paucity of literature on MSM and HIV, particularly in the context of COVID-19. The study will also inform future programming for MSM and HIV. Furthermore, the researchers felt that the study was significant since the MSM community is difficult to access. In the given context, the aim of this study was to gain insights into experiences of MSM with regard to accessing healthcare services during the COVID-19 lockdown period in Zimbabwe.

## Materials and methods

### Study design

An interpretative phenomenological analysis (IPA), which is one of the designs in qualitative approach, was chosen to gain an insight into the lived experiences of MSM in the COVID-19 era. The researchers are clinicians and published researchers on HIV and key populations. This background and experience enhanced the researchers' understanding of the phenomenon under study. The IPA enables the researchers to understand the innermost deliberation of the 'lived experiences of study participants and how they make sense of their major life experiences [23, 24]. In addition, the approach allows study participants to narrate their stories through their lived experiences [25, 26]. The approach encourages researchers to adopt an insider perspective, to explore and ask critical questions about the things the study participants say so as to gain an in-depth understanding of their experiences [24]. Furthermore, the reporting of the study was guided by the Standards for Reporting Qualitative Research [25].

### Study setting

In Zimbabwe same sex sexual conduct is criminalised. The discriminatory environment makes it difficult to provide person-centred HIV care services targeting specifically the MSM community [21]. Zimbabwe also has a volatile economy and is politically unstable. In addition to facing stigma and discrimination, the general acceptance and acknowledgement of key

populations by the government, health workers, and the public is still limited. The COVID-19 pandemic aggravated the challenges faced by the MSM community in accessing HIV care services.

The study was carried out in Bulawayo the second largest city of Zimbabwe. Bulawayo has two central hospitals, 17 City health facilities and 2 private facilities that provide HIV care to key populations. Most care providers in Zimbabwe do not have the appropriate capacity to work with and offer services to key populations. In an effort to enhance access to HIV care services, 17 city health facilities had some of their providers trained in key populations friendliness. Two of these facilities provide a minimum package for key population-friendly service provision in the public health care system. The emergence of COVID-19 disrupted these efforts in increasing access to service provision.

## Ethical considerations

Ethical consideration and approval were obtained from the Medical Research Council of Zimbabwe (Ethics Clearance Number MRCZ/A/2659). The research was conducted as per the Declaration of Helsinki. All participants gave written informed consent and permission to audio tape the interviews. The transcripts were then transcribed verbatim, and the data coded. The interview data and the key codes were locked in a password protected computer.

## Sampling

To gain access this population that is difficult to access (namely, MSM), one of the organisations that provide services for key populations was requested to assist with the recruitment process. Purposive sampling augmented with snowballing was used to identify potential study participants through the networks of those participants who had already been interviewed [26]. The first two participants were accessed as referrals from the staff working at an organisation that provides services to key populations after being told about the study. Thereafter potential study participants who were willing to participate in the study, had their contact details shared with the researchers to facilitate the recruitment process. The inclusion criteria for participating in the study were as follows: self-identifying as men who have sex with other men, aged 18 years or older and having been receiving ART or pre-exposure prophylaxis (PrEP) from any of the healthcare facilities in Bulawayo before COVID-19 lockdown. A total of 14 participants that met the inclusion criteria eventually participated in the study. The sample size was determined by saturation, which refers to the point at which the data collection process fails to yield new information relevant to the study [27]. Because of the in-depth nature of the interviews undertaken, the sample size of fourteen is justifiable in a phenomenological study as supported by Polit and Beck [26] who reported a sample of ten or more as adequate. This is corroborated by Smith and Osborn [24] who posit that a sample of six to twenty is suitable for a phenomenological study.

## Data collection

\Data were collected between November 2020 and February 2021. This time period gave the researchers an opportunity to conduct follow-up interviews in case any information needed to be verified. The primary concern of IPA researchers is to elicit rich, detailed, and first-person accounts of experiences and phenomena under study [24]. In keeping with IPA principles, data collection was done using semi-structured interview guides. All interviews were in-depth, one-on-one interviews and began with the following central question: Please describe your experiences concerning accessing HIV prevention (PrEP) services or HIV treatment (ART) during the COVID-19 lockdown period. Apart from the central research question, probing

questions such as the following were asked to allow the participants to elaborate on or clarify statements: From your point of view, was it easy or difficult to access HIV prevention, treatment, and care services? Why? Then what happened? What were your feelings/emotions like? IPA interviews are meant to open up and develop a relationship with the participants so that their 'lived experiences' can be explored and analysed [24]. Therefore, probes and prompts were used to elicit more detailed information from participants [24, 28]. An audio recording was made of every interview and every interview lasted 60 minutes or more. In addition, field notes were also utilised to capture and describe non-verbal cues observed during the interviews. A reflexive diary was utilised to ensure that all questions in subsequent interviews were open ended. A pilot study that involved four MSM clients was also conducted in one of the centres that provide HIV services for key populations. This process helped the researchers to refine the research instrument and to rephrase some of the follow-up questions. In all the interviews the researcher ensured that the purpose of the study was well explained to the participants. A written informed consent was also obtained from each participant before proceeding with the data collection and audio recording of the interview. The research participants were encouraged to keep all information shared during the interview sessions confidential.

## Data analysis

All audio-recorded interview data were transcribed verbatim into written texts. The transcripts were analysed independently by two researchers, using the Interpretative phenomenological analysis (IPA) framework. The IPA is an approach that aims to provide detailed examinations of personal lived experience [24]. The researchers read each transcript several times and listened to the audio recording a few times and made notes in the process, which helped them to immerse themselves in the data and to recall the atmosphere and the setting of the interview. The researchers made notes on their observations and reflections about the interview experience. All researchers, working independently, transcribed the audio recordings verbatim and translated them from Ndebele into English. The two researchers who collected data, and the third one who acted as an independent coder, conducted open coding of each transcript. The following steps, as outlined by Smith and Osborn [15], were followed: (1) reading and rereading all the transcripts, (2) taking notes and developing emergent themes, (3) clustering the emergent themes, (4) crafting a master table of themes composed of superordinate themes, subthemes and extracts from the interviews, (5) examining and comparing the similarities between the master tables of the themes and (6) compiling a single master list composed of a superordinate theme, themes and subthemes. Following this process, the research team met to compare and discuss their respective master tables of themes. The team had a consensus discussion and agreed on the final master table, composed of superordinate themes, subthemes, and associated excerpts from transcripts [23, 29].

## Measures of trustworthiness

The data was evaluated as per Guba and Lincon criteria to determine the reliability and validity of the results [30]. Open ended questions were used to ensure that the researchers' description and interpretation accurately captured the experiences of the participants. To enhance authenticity, verbatim extracts from the interviews were utilised and member checking was done [31]. The process of member checking was achieved by providing the participants with research findings as well as discussing the results with them. Data saturation was reached at client number 12, but collection continued up to 14 participants. The study utilised the IPA framework for analysis outlined by Smith and Osborn [24]. This ensured auditability and

consistency of the process of data collection. Extracts of participants' views are also provided. To ensure neutrality, the researchers suspended their biases through bracketing [27].

## Findings

### Biographic information

The sample consisted of 14 participants who were aged between 19 and 45 years and who self-identified as MSM. Of these, seven participants were on PrEP, and the remainder (seven) were on ART. The study participants accessed services from six facilities, four of which were public sector facilities and two were privately owned. The study participants were clients who accessed either PrEP or ART in health-care facilities in the city of Bulawayo, Zimbabwe. None of the study participants were married or formally employed. Most of the participants defined their sexual orientation during the interviews as gay while only two indicated that they were bisexual. The term sexual orientation refers to the multidimensional consisting of individuals' sexual identities, attractions, and behaviours [32, 33]. On the other, hand the term bisexual is associated with being attracted to both men and women while in some instances the individuals also date trans people and hence there is no one way to be bisexual [33]. The biographic data of the study participants are presented in Table 1 below:

Two super-ordinate themes were formed during the analysis as follows: barriers to accessing HIV prevention, treatment, and care services (T1) and psychosocial and economic effects of COVID-19 (T2). The two super-ordinate themes overarched six themes and eleven sub-ordinate-themes (Table 2).

### 1. Barriers to accessing HIV prevention, care and treatment services (T1)

This superordinate theme relates to barriers experienced by MSM when assessing HIV services. This superordinate theme is composed of the following themes: police roadblocks and lack of access to service delivery.

### 1.1. Police roadblocks

The police roadblocks that were part of the government's COVID-19 response had had a profound impact on clients' attempts to access HIV prevention and treatment services.

**Table 1. Demographic and service-related characteristics of study participants.**

| Interviewee code | Age range in years | Sexual Orientation | Level of education | Employment status | Service | Duration on PrEP/ART |
|---|---|---|---|---|---|---|
| M001 | 25–30 years | Gay | Tertiary | Plumber | PrEP | 2 years |
| M002 | 25–30 years | Gay | Tertiary | Teacher | PrEP | 1 year |
| M003 | 18–24 years | Gay | High School | Unemployed | PrEP | 3 years |
| M004 | 25–30 years | Gay | Secondary School | Vegetable vendor | ART | 20 months |
| M005 | 31–35 years | Bisexual | Secondary School | Sex worker | PrEP | 3 years |
| M006 | 40–45 years | Gay | Secondary School | Shop owner | ART | 4 years |
| M007 | 25–30 years | Gay | Secondary School | Gardener | PrEP | 1 year |
| M008 | 25–30 years | Gay | Secondary School | Unemployed | PrEP | 13 months |
| M009 | 31–35 years | Gay | Secondary School | Farmer | ART | 3 years |
| M0010 | 18–24 years | Gay | Tertiary | Domestic worker | ART | 2 years |
| M0011 | 31–35 years | Gay | Secondary School | Sex worker | ART | 2 years |
| M0012 | 25–30 years | Gay | Secondary School | Sex worker | PrEP | 1 year |
| M0013 | 25–30 years | Gay | Secondary School | Sex worker | ART | 2 years |
| M0014 | 31–35 years | Bisexual | Secondary School | Community worker | ART | 18 months |

**Table 2. Superordinate, theme, and sub-themes on MSM and COVID-19.**

| Superordinate Theme | Theme | Sub-theme |
|---|---|---|
| Barriers to accessing HIV prevention, treatment and care services (T1) | 1.1. Police roadblocks | 1.1.1, Demand for travel authorisation letters |
| | | 1.1.2. Treatment interruptions and infringing into clients' privacy. |
| | 1.2. Lack of access to service delivery | 1.2.1. Differentiated service delivery in the community |
| | | 1.2.2. Effects of adapting service delivery in health care facilities |
| Psychosocial and economic effects of COVID-19 (T2) | 2.1. Loss of income | 2.1.1. Changes in the business model |
| | | 2.1.2Changes in lifestyle |
| | 2.2. Intimate Partner Violence | 2.2.1. Physical and verbal abuse |
| | | 2.2.2. Emotional effects of Intimate Partner Violence |
| | 2.3. Psychological effects | 2.3.1. Unintended disclosure of HIV status |
| | | 2.3.2. Physical and emotional effects |
| | 2.4. Availability or lack of a support system | 2.4.1. Family or peer support |
| | | 2.4.2. Professional Support |

**1.1.1. Demand for travel authorisation letters.** Clients felt stigmatised and embarrassed in roadblocks and were forced to take different routes. The following extracts demonstrate the pervasive impact of roadblocks on clients:

'Due to the lockdown, I was required to produce evidence of a medical record for me to pass at the roadblock. I had to wake up early, pass through five roadblocks for me to get to the clinic. The time spent on the road was too much since the only means of transport was ZUPCO buses that took long to get from one destination to the next.'

(M013)

'It was difficult to go and collect medication. I was turned away by the police and ended up with no ARV medicines. I used to request ARVs from a friend. I was only able to pass through the roadblock after showing them the empty ARV bottle as proof that I really don't have the medicines.'

(M010)

**1.1.2. Treatment interruptions and infringement on clients' privacy.** Clients also felt their privacy had been interfered with and indicated that there had to discontinue taking PrEP as the following excerpts show:

'At the police roadblock, the police were asking me in public what PrEP is used for. I felt angry that they were interfering with my privacy. They further asked me why I was taking PrEP, and whether I was a sex worker since PrEP is taken by sex workers.'

(M008)

'I made efforts to use other roads and avoid the police roadblocks, but I failed. I ended up going more than two weeks without PrEP until I got PrEP refills from a peer educator.'

(M003)

## 1.2. Lack of access to service delivery

The COVID-19 restrictive measures led to reduced access to service delivery. To ensure continuity in HIV care adjustment in service delivery was done.

**1.2.1. Differentiated service delivery in the community.** In response to the restrictive measures implemented to deal with COVID-19, ensure continuity of care, community differentiated service delivery approaches in HIV care were scaled up. Some clients appreciated the use of this approach, as exemplified by the following extract:

'Key Populations Peer Educators provided us with PrEP supplies in the community. I felt good that services were brought to us in the community since transport was a challenge. They would phone and we met at a convenient place away from home.'

(M003)

'I was given a six-month supply of ARVs and food hampers closer to my home. I appreciated the community refills as it eased challenges associated with travelling during the lockdown.'

(M004)

**1.2.2. Effects of adapting service delivery in health care facilities.** Whilst the clients appreciated the innovative strategies implemented to ensure service continuation, they noticed differences between the adapted services and the usual services, as demonstrated by the extract below:

'There were fewer members of staff at the clinic where I collect my ARVs. We went through the COVID-19 screening process, we were served fast and given a six months' supply. I noted that the usual service had changed as there was inadequate time for provider–client interaction. It was difficult to express myself during the counselling session because the counsellor seemed anxious and in a rush.'

(M010)

## 2. Psychosocial and economic effects of COVID-19 (T2)

This superordinate theme is composed of the following themes: loss of income, intimate partner violence, psychological effects and availability and lack of social support.

## 2.1 Loss of income

The study found that during the COVID-19 lockdown, clients had experienced a loss of income that affected their lifestyles and the way they operated.

**2.1.1. Changes in the business model.** The loss of income resulted in changes in the business model as shown below:

'Before the lockdown, I had a lot of clients. I was able to select clients from the bars. With the lockdown, it was difficult to get money for food and I resorted to looking for clients through social media.'

(004)

'Income was a challenge, business could not continue, I could not continue paying rentals at my shop. Therefore, I surrendered the shop I rented to the owner.'

(M012)

**2.1.2. Changes in lifestyle.**    The loss of income resulted in lifestyle changes as illustrated by the following extracts:

'I am a hustler; I was always able to sustain myself. But, with the advent of COVID-19, my income dwindled, forcing me to go back home, where both my parents had lost jobs due to COVID-19. My partner had to chip in with groceries.'

(M002)

Loss of revenue led to life adjustments, as the following extracts show:

'As a male sex worker, because of COVID-19 restrictions with bars closed, I lost most of my clients and some of my clients lost their jobs and hence could not assist me. I could not get additional benefits such as food, toiletries and money for rentals.'

(M005)

'. . . usually staying on my own, business not doing well, I was forced to go back home, now found myself stuck at home.'

(M001)

## 2.2. Intimate partner violence

**2.2.1. Physical and verbal abuse.**    Because of the lockdown and restricted movement, MSM and their partners were in a sense locked up, generating tension, which in some cases resulted in violence. One participant remarked as follows in this regard:

'I had an experience of intimate partner violence more than once. My partner was trying to take control of my life since he paid for my rentals. There were instances of physical fights and verbal abuse as he accused me of being too demanding.'

(M005)

**2.2.2. Emotional effects of intimate partner violence.**    Another client had this to say:

'I used to fight with my partner, and he was verbally aggressive. This affected me emotionally as I was dependent on him as the breadwinner.'

(M004)

'I used to stay with my partner. He was very harsh, and this traumatised me. I reported him to the police.'

(M010)

## 2.3. Psychological effects

The restrictions in movement had a psychological impact on many clients.

**2.3.1. Unintended disclosure of HIV status.** The police roadblocks were intimidating, inducing a sense of fear and anxiety, which was aggravated by breaches of confidentiality, as shown by this extract:

'At a police roadblock I was forced to show the police my medical records. They were reading, checking and asking me about my medical records, that showed that I was taking ARVs. I felt violated.'

(M013)

Restrictive measures required travellers to produce authorisation letters, which some ART clients did not have, in lieu of which they produced ART records. This resulted in unintended disclosure, which affected the clients emotionally, as illustrated by the following extracts:

'I felt my privacy was interfered with, having to expose my ARV containers, I felt angry, people in the bus seeing that I am on ART and some of them were neighbours.'

(M011)

'This experience of having to disclose my HIV status to the police (strangers) pained me so much because it happened in the presence of other people. I felt stigmatised and emotionally pained.'

(M014)

**2.3.2. Physical and emotional effects.** Sometimes MSM experience anxiety about how to take their medication in privacy due to COVID-19 adjustments. In addition, some had lost employment. The following participants commented as follows in this respect:

'I felt anxious, depressed and at times suicidal.'

(M012)

'The whole lockdown stressed me. I ended up having persistent headaches, went to the hospital and I was told that I was suffering from depression. I could not cope socially and financially.'

(M007)

'. . .things changed overnight, I was unable to pay my rent and buy food. The situation called for an adjustment in life. I felt anxious, depressed and at times felt suicidal because of having to change my lifestyle.'

(M006)

## 2.4. Availability or lack of a support system

**2.4.1. Family or peer support.** Clients got support through virtual means like WhatsApp or from family members or colleagues. The following extract illustrates these two factors:

'For my concerns, I spoke to my partner, mum, friends from the KP community and this helped me so much. I also got assistance for rentals and food supplies from my brother. I was not able to meet my with my partner, we would talk on the phone and on WhatsApp.'

(M002)

Whilst some clients experienced intimate partner violence, others had a different experience, as shown by the following extract:

'During the lockdown, I had an opportunity to bond with my partner. I learnt that he cares for me.'

(M009)

**2.4.2. Professional support.** 'My main support system was the counsellor from . . . . . . [an institution that caters for key population community], [who] was always available on standby to provide support.'

(M007)

## Discussion

Findings from this study reveal that COVID-19 and the efforts taken to curb its spread have caused interruptions in HIV service provision. The study established that study participants had had negative encounters at police roadblocks in their efforts to travel to treatment centres during the pandemic. An American study [34] highlights that substantial proportions of MSM clients were already reporting barriers associated with accessing HIV prevention and treatment services and if the COVID-19 pandemic continues, access to these vital services may become a great challenge [34]. In this study, participants often felt their privacy was violated as their medical records were scrutinised and intrusive questions were asked by law enforcement officials. These findings are not unique to Zimbabwe. Similar experiences of key populations have been reported on in other studies. One study, for example, reporting on cross-cutting issues on HIV and key populations: found that this group had been victims of punitive crack-downs and raids of their homes in different countries [1, 5]. MSM in China also faced barriers in accessing treatment during the COVID-19 induced lockdown [22]. According to a newspaper report [35], obtaining letters authorising people's movement to treatment facilities and allowing them to pass police roadblocks was difficult. In Zimbabwe, as in Uganda, the authorised public transport often experienced delays and took long to get to the treatment centres. Under these circumstances, it is advisable to scale up telehealth options, such as telephonic services and other eHealth platforms, to facilitate the provision of services in a safe and flexible manner, so as to avert disruptions in treatment [1, 34]. Moreover, law enforcement agents should be trained in more humane methods of policing (sensitivity to individual needs of members of the public). For example, the intrusive nature of the questions at the police roadblocks could have been avoided. The delivery of HIV prevention, care and treatment services should make provision for the mental wellbeing of MSM [36].

The study found that the participants had adjusted to the new reality by means of the multi-month dispensing of antiretroviral therapy and community refills of PrEP. In some settings, like the one in this study, health facilities were able to provide ART services through the utilisation of differentiated service delivery approaches [10, 37]. Expedited differentiated service

delivery in respect of HIV during the COVID-19 pandemic is commendable [1, 2]. This is a patient-centred approach that simplifies and adopts HIV service delivery across the cascade of services to serve the needs of the people [38]. The differentiated service delivery is critical for maintaining gains achieved in HIV epidermic control [10]. Telehealth and multi-month dispensing are among the innovations that the study found to be utilised to facilitate treatment continuum. The use of peer support systems and home visits, as noted in the study in the context of COVID-19, is in keeping with WHO recommendations [1].

The study also found that some participants had experienced disruptions to and interruptions in ART and PrEP service delivery due to travel restrictions and police roadblocks as part of COVID-19 lockdown measures. Other studies indicate that similar disruptions occurred during COVID-19 lockdowns in South Africa [39], Uganda [40] and Kenya [41–43]. In respect of a study in China [28] face-to-face consultations and laboratory testing have been reduced because of the COVID-19 pandemic and its impact on HIV care, and lockdown restrictions affect people's ability to access testing or collect medicine. The authors further posit that the resultant gaps in HIV treatment and care could result in further HIV transmission or deaths.

The study found that those who were hustlers or in the informal sector also lost income. This finding was not unique to the study setting, as studies conducted elsewhere illustrate [44]. Due to the COVID-19 pandemic, the majority of key populations lost their jobs [44]. In addition, COVID-19 lockdown restrictions saw the closure of bars and night clubs where MSM used to get their clients, leading to a loss of revenue. The loss of jobs and income forced some of the participants to go back home and stay with their parents or relatives, many of whom were homophobic [44, 45]. Young workers were hard hit by job losses in restaurants and the service sector [46]. Among these young workers were MSM. Apart from the unemployment and underemployment resulting from COVID-19, the study found that COVID-19 induced anxiety and worry in MSM. The loss of income experienced by participants in this study was similar to what happened to their Kenyan counterparts, where the loss of jobs and income among MSM during the COVID-19 lockdown resulted in financial and food insecurity [42].

In this study, it was evident that participants had experienced intimate partner violence (IPV) during the COVID-19 pandemic. Several studies have shown that this phenomenon affects the continuity of HIV care [47, 48]. Moreover, several reports show that restrictive measures imposed to deal with COVID-19 have resulted in an increased incidence of IPV. This violence affects the uptake and adherence to HIV treatment and care. In addition, due to the stay-at-home requirements, many members of the lesbian, gay, bisexual and transgender (LGBT) community experienced IPV [46]. According to a study [49], evidence from the experiences of those affected by IPV will assist inform intervention and guidance to programming during pandemics or times of crisis.

The IPV noted in this study was also observed in several other studies [50, 51]. Furthermore, another study [52] argues that financial dependence on the abuser makes it difficult for the victim to leave. We noted that many participants in the study had lost their source of income. Studies contend that key populations' loss of income and residential instability are aggravated when they do not have the resources to physically distance themselves from their abusers, do not have access to safety nets or do not have the option to work from home [11, 12].

The unemployment and underemployment caused by the COVID-10 pandemic contributed to anxiety and worry, which were aggravated by feelings of abandonment and financial stress after job losses [51]. The continued COVID-19 lockdown measures, coupled with mandatory social distancing and other isolation measures, restrict people's access to essentials such as food, medicine and transport, thus straining their mental wellbeing [53]. This affects them psychologically, impacting on their quality of life negatively. The impact of restrictive

measures manifests itself in the mental and social spheres [54]. A global survey conducted by Greenhalgh [55] among MSM demonstrated high levels of loneliness and depression due to restrictive measures imposed as a result of COVID-19. The impact of the restrictive measures was manifested negatively in the psychological and mental wellbeing of MSM. The rupture in socio-affective networks owing to the COVID-19 pandemic, specifically not being able to meet friends or interact in groups and not having access to nightlife activities, pose a challenge to MSM [56].

UNICEF notes how the psychological impact of COVID-19 further exacerbates the marginalisation of young key populations [57]. Similarly, this study found that police roadblocks induced fear and anxiety in MSM who commuted to access HIV care services. The study also observed that under normal circumstances, people living with HIV and MSM are at risk of mental health challenges, which has been further aggravated by the COVID-19 pandemic. This emphasises the need for psychosocial support for MSM. The United Nations Population Fund (UNPFA) cites the Director of the Youth Advocates Zimbabwe (YAZ) as calling for a help line for key populations so that they are provided with a customised service [58].

In this study, participants expressed gratitude for the food parcels distributed by civil society during the COVID-19 lockdown, especially in view of the restriction on movement coupled with job losses for some of them. A similar trend was noted by ILGA World [59]. In South Africa, civil society organisations like the Scalabrin Centre assisted in distributing food parcels to LGBT migrants, and Femme Projects helped shelters with food and toiletries. The study also found that the participants had been supported by peers through virtual means like WhatsApp or scheduled calls. This finding is consistent with findings by UNICEF East Asia and Pacific that young MSM used online channels to access peer support [57]. Similarly, the United Nations Population Fund (UNFPA) reports that there was a youth help line in Zimbabwe for key populations, which was complementary to existing static clinics [58]. The help line had a good referral network and linked key populations to different service providers. UNAIDS also report that key populations were supported by the provision of peer navigation through virtual means and the existence of drop-in centres [50]. Moreover, the UNFPA procured and distributed personal protective equipment for key populations at dedicated clinics and drop-in centres [58].

## Strengths and limitations

This study is one of the pioneering endeavours in trying to understand the impact of COVID-19 on men who have sex with men accessing HIV care services. It highlights the need for specific targeted interventions to facilitate treatment continuity for the MSM client even under COVID-19 conditions. However, the findings are preliminary and highlight the need for further investigations on a broader scale.

The limitation for this study is that because of restrictions in movement, some participants might have been excluded. The researchers also acknowledge the possibility of sampling bias that could have been occasioned by the use of snowballing sampling. In spite of this, the study reflects the experiences of the MSM community. Therefore, the study contributes valuable evidence necessary for programme planning.

## Conclusion

The study highlights the negative impact of COVID-19 on MSM in respect of their access to HIV prevention, treatment and care services, as well as its psychosocial and economic effects. The findings highlight the manner in which COVID-19 may widen social inequities and health disparities since COVID-19 has a disproportionately negative effect on the MSM community

and other marginalised and vulnerable groups. Continued monitoring of the impact of the COVID-19 pandemic on MSM is critical in order to gain an appreciation of the emerging and long-term effects of the pandemic. This is to ensure our responses are adapted to sustain the gains made toward increasing access to HIV and promoting HIV epidemic control. Based on the results, with regards to depression, anxiety and intimate partner violence, scaling up tele-health where a mental health counsellor/psychologist would provide counselling services online is recommended. In that regard, the closure of MSM clinics noted in a study in Zimbabwe [4] and Kenya [3] was not a good idea. Keeping the clinics open would have ensured treatment continuity. Government and non-governmental organisations to set up a fund for income generation projects to mitigate loss of income for this vulnerable group. A similar fund was launched in 2020, piloted in Brazil, Ghana, India, Madagascar and Uganda [60].

## Supporting information

**S1 File. Standards for Reporting Qualitative Research (SRQR).**
(ZIP)

**S2 File. Superordinate themes, themes, sub-themes and sample quotes of MSM experiences during COVID19.**
(TIF)

**S3 File. Interview Guide.**
(TIF)

**S4 File. IPA procedure.**
(TIF)

## Acknowledgments

The authors acknowledge and thank all the study participants and the organisation that assisted with the recruitment of participants.

## Author Contributions

**Conceptualization:** Idah Moyo, Azwihangwisi Hellen Mavhandu-Mudzusi, Musa Sibindi.

**Data curation:** Idah Moyo, Freddy Lipeleke.

**Formal analysis:** Idah Moyo, Azwihangwisi Hellen Mavhandu-Mudzusi, Freddy Lipeleke.

**Investigation:** Freddy Lipeleke, Musa Sibindi.

**Methodology:** Idah Moyo, Azwihangwisi Hellen Mavhandu-Mudzusi, Freddy Lipeleke, Musa Sibindi.

**Project administration:** Idah Moyo, Azwihangwisi Hellen Mavhandu-Mudzusi.

**Supervision:** Idah Moyo, Azwihangwisi Hellen Mavhandu-Mudzusi.

**Visualization:** Freddy Lipeleke, Musa Sibindi.

**Writing – original draft:** Idah Moyo, Freddy Lipeleke, Musa Sibindi.

**Writing – review & editing:** Idah Moyo, Azwihangwisi Hellen Mavhandu-Mudzusi.

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
