## [Decision Letter · Decision Letter 0]

12 Apr 2022

PONE-D-21-18907The experiences of men who have sex with men in accessing HIV prevention, treatment, and care services during COVID-19 in ZimbabwePLOS ONE

Dear Dr. Moyo,

Thank you for submitting your manuscript to PLOS ONE. After careful consideration, we feel that it has merit but does not fully meet PLOS ONE’s publication criteria as it currently stands. Therefore, we invite you to submit a revised version of the manuscript that addresses the points raised during the review process.

The reviewers raised a number of concerns about the presentation of your data and results, the description of your methodology and the rationale for your study. Their comments can be viewed in full below, and in the attached file.

We look forward to receiving your revised manuscript.

Kind regards,

Natasha McDonald, PhD

Associate Editor

PLOS ONE

Journal Requirements:

2. When reporting the results of qualitative research, we suggest consulting the COREQ guidelines  or other relevant checklists listed by the Equator Network, such as the SRQR, to ensure complete reporting (http://journals.plos.org/plosone/s/submission-guidelines#loc-qualitative-research). Moreover, please provide the interview guide used as a Supplementary File.

4. We note you have included a table to which you do not refer in the text of your manuscript. Please ensure that you refer to Table 1 in your text; if accepted, production will need this reference to link the reader to the Table.

Reviewers' comments:

Reviewer's Responses to Questions

**Comments to the Author**

1. Is the manuscript technically sound, and do the data support the conclusions?

Reviewer #1: Yes

Reviewer #2: Yes

2. Has the statistical analysis been performed appropriately and rigorously? 

Reviewer #1: Yes

Reviewer #2: N/A

3. Have the authors made all data underlying the findings in their manuscript fully available?

Reviewer #1: No

Reviewer #2: No

4. Is the manuscript presented in an intelligible fashion and written in standard English?

Reviewer #1: Yes

Reviewer #2: Yes

5. Review Comments to the Author

Reviewer #1: This is a well written manuscript detailing The experiences of men who have sex with men in accessing HIV prevention, treatment, and care services during COVID-19 in Zimbabwe. A few comments

1. Line 179-181, the statements ‘Therefore, probes and prompts were used to elicit more 180 detailed information from participants [19]. Therefore, probes and prompts were used to elicit more detailed information from participants [15,19]’

are repetitive.

2. Table1, duration on what? It would be better to clearly state on the subtitle

3. Line 255, did you intended to say …. Had had…., cross check

Reviewer #2: This paper addresses the important perspective of MSM in Zimbabwe accessing PrEP or ART. The quotes are informative and highlight the experience of MSM during COVID.

The title should mention that this is a qualitative analysis

ABSTRACT

In the methods section of the abstract, the authors should mention the data collection methods (ie we conducted in depth interviews with …) before the data analysis methods (IPA design)

“Results showed several barriers to accessing HIV services by MSM that warranted

adjustments to service delivery.”

This sentence should be more specific as it does not provide substantial information

“The study also found that COVID-19 and related restrictive measures had psychosocial and economic effects that included loss of income, intimate partner violence and psychological effects.”

This sentence could also be more specific.’

Conclusions suggest community-based HIV services but this is not mentioned in the results.

INTRODUCTION

“These measures, though well-meant, have had a negative impact on HIV services, resulting in challenges and risks to the infected and affected groups”

This sentence makes it seems as thought COVID restrictions were misguided although “well meant”. Do the authors think lockdowns were not necessary to prevent deaths due to COVID? If not, I suggest removing “tough” and “well meant” when talking about COVID protocols. The rest of this sentence is vague and provides no information about how COVID has affected key populations. Finally, I would avoid using “the infected” to talk about people with HIV.

“The Joint United Nations Programme on HIV/AIDS (UNAIDS) [2] indicates that key populations are particularly vulnerable to HIV service interruptions.”

Why is this the case? What kind of disruptions have occurred?

“The emergence of COVID-19, together with the associated national lockdowns and restricted movement, has brought diverse challenges that further aggravate the situation for key populations.”

This sentence is vague and provides no information. What are the “diverse challenges”?

“According to UNAIDS [2], the COVID-19

84 pandemic has affected everyone, inclusive of key populations, who are at a

85 higher risk of acquiring HIV.”

This sentence is vague

“The Joint United Nations Programme on HIV/AIDS indicates that the gains that have been made in implementing HIV programmes and increasing access to HIV services for key populations are at risk of being

reversed as a result of disruptions caused by COVID-19”

This sentence is vague.

“To curb the spread of COVID-19, the World Health Organization (WHO) called for adjustments in HIV care service delivery [1].”

What adjustments? This sentence is vague.

The introduction needs to be completely re-written. It provides no information about the impact of COVID-19 in Zimbabwe, the characteristics and health seeking behavior of MSM in Zimbabwe, previous research on MSM’s engagement and experience with HIV services, gaps in the research and the need for the present study,. There is insufficient description of the objectives of the current study (“other than explore the experience…”) and the introduction is too long.

METHODS

Design—the IPA analysis should go in the analysis and not design section

Ethical considerations

This section should be greatly shortened

Sampling:

“Purposive sampling facilitated selection of participants by virtue of their capacity to provide richly textured information, relevant to the phenomenon under study.”

This sentence in unclear

The sample size is very small (14 people). This should be mentioned in the limitations.

Data collection

It would be helpful to include the interview guide as an appendix

Interview guide: “From your point of view, what are the factors that facilitated or impeded your access to HIV prevention, treatment and care services?”

This seems like a hard question

Data analysis

“Interpretative phenomenological analysis (IPA), an approach which aims to provide detailed examinations of personal lived experience [15].”

This sentence is repetitive and is mentioned several times throughout the paper.

Measures of trustworthiness

The purpose of this section is unclear.

“A reflective diary was kept ensuring neutrality and objectivity.”

The meaning of this is unclear

RESULTS:

Table one should be a summary of demographics and not a list of every single participant

The quotes from the participants are very informative

Loss of income and change in lifestyle seems like they could be combined into one theme

Are there quotes to illustrate the difference?

CONCLUSION

The authors state “Moreover, law enforcement agents should be trained in more humane methods of policing”

Can you provide examples of how this can be done or prior interventions in the literature that have been successful

Can the authors suggest potential interventions to address the barriers and consequencese of COVID observed in this study? Eg, loss of employment/income, IPV, depression/anxiety due to COVID and the lockdowns

One limitation to note in the “limitations” section is that it’s not possible to assess generalizability when using snowball sampling.

Are the negative experiences noted here specific to MSM or are they more generally experienced in the community due to COVID? How are the experiences of MSM distinct from the general community?

6. PLOS authors have the option to publish the peer review history of their article (what does this mean?). If published, this will include your full peer review and any attached files.

Reviewer #1: No

Reviewer #2: No

---

## [Author Response · Author response to Decision Letter 0]

25 May 2022

File attached showing how authors responded to the reviewers' comments

---

## [Decision Letter · Decision Letter 1]

19 Jul 2022

PONE-D-21-18907R1The experiences of men who have sex with men in accessing HIV prevention, treatment, and care services during COVID-19 in Zimbabwe, An interpretive phenomenological analysis studyPLOS ONE

Dear Dr. Moyo

Thank you for submitting your manuscript to PLOS ONE. After careful consideration, we feel that it has merit but does not fully meet PLOS ONE’s publication criteria as it currently stands. Therefore, we invite you to submit a revised version of the manuscript that addresses the points raised during the review process.

We look forward to receiving your revised manuscript.

Kind regards,

Yogan Pillay, Phd

Academic Editor

PLOS ONE

Journal Requirements:

Additional Editor Comments (if provided):

As above

Reviewers' comments:

Reviewer's Responses to Questions

**Comments to the Author**

1. If the authors have adequately addressed your comments raised in a previous round of review and you feel that this manuscript is now acceptable for publication, you may indicate that here to bypass the “Comments to the Author” section, enter your conflict of interest statement in the “Confidential to Editor” section, and submit your "Accept" recommendation.

Reviewer #3: (No Response)

2. Is the manuscript technically sound, and do the data support the conclusions?

Reviewer #3: Yes

3. Has the statistical analysis been performed appropriately and rigorously? 

Reviewer #3: N/A

4. Have the authors made all data underlying the findings in their manuscript fully available?

Reviewer #3: Yes

5. Is the manuscript presented in an intelligible fashion and written in standard English?

Reviewer #3: Yes

6. Review Comments to the Author

Reviewer #3: This is a well presented manuscript r on a very important and historically neglected area of research. I commend the authors for carrying out this important work.

The manuscript adequately addressed the comments of Reviewers 1 and 2 in the first round of peer-review. However, there are still notable issues, most minor and some more substantial, before the paper is publishable.

1. The most substantial issue is Table 1 on page 14. Why are there two columns differentiating between sexual orientation (all listed as gay) and sexuality (all listed as homosexual)? This does not make conceptual or practical sense for two reasons. Firstly, 'homosexual' is generally avoided, due to its history as an overly medicalised term, and is now outdated. Secondly, there is no difference between 'gay' and 'homosexual' - what do the authors mean to imply here?

2. Related to the above issue is whether this is indeed a study of MSM at all, or a study on gay men in Zimbabwe? There are conceptual differences. The term "MSM" and "key populations" belongs to public health discourse and do not necessarily account for the actual identity of the people it homogenises under the label "MSM". A person listed as MSM might not be gay; they may be heterosexual or bisexual, etc. In your study, ALL the participants were listed as "gay" - how was this information given? Please provide the manner in which the demographic variables were obtained so that it is clear whether the participants were given pre-determined options for their 'sexual orientation' and 'sexuality' or whether there were opportunities for them to self-identify differently. If indeed all the participants IDENTIFIED as gay, then why was the study framed as a study of MSM and not as a study of gay men? Do you have evidence that the participants themselves prefer the term MSM? Where they asked?

The authors can turn to the following reference for further clarity: Young RM, Meyer IH. (2005) The trouble with “MSM” and “WSW”: Erasure of the sexual-minority person in public health discourse. American Journal of Public Health, 95(7):1144–9.

In my opinion, the above two issues must be clarified before the paper is accepted for publication.

Other minor issues include the following:

3. A few grammar errors, e.g. line 27, line 82

4. The uncritical use of the phrase "hard to reach" on line 138 (there is a lot of critique around this term)

5. Remove "best opportunity" on line 146 as many methods can achieve your research goals.

6. Line 153 you say "the study was performed" according to SRQR; however, the SRQR is not meant to dictate how to perform the actual study. It guides the reporting of the subsequent manuscript.

7. The SRQR S.6 requires you to comment on Researcher Characteristics. There is almost no reflexivity on the characteristics or positionality of the authors, in relation the participants. More detail is needed here: who are you, why did you personally want to research this area, what previous engagements have you had with similar participants, are any of your members of these communities? etc.

8. Line 162 - what were the incentives to participate in this study? It is unclear why people chose to participate in a study that is potentially stigmatising to their identity/

9. Line 165 - "KP friendliness" - you have used the acronym "KP" before.

10. Line 168 - You note that participants were selected "by virtue of their capacity to provide richly textured information". How did you know beforehand whether or not a potential participant would be able to provide "richly textured information"?

11. Line 224-5 - You changed the text in the manuscript but surely you would have already asked the question based on the original phrasing?

12. On Table 1, an international audience may not know what O-Level and A-Level are.

13. On Table 1, under employment status, is there a difference between Self-Employed and 'Male Sex Worker'? Would a Male Sex Worker not also be self-employed in some instances?

I hope that my comments help strengthen the manuscript for potential publication. All the best to the authors!

7. PLOS authors have the option to publish the peer review history of their article (what does this mean?). If published, this will include your full peer review and any attached files.

Reviewer #3: **Yes: **Suntosh R. Pillay

---

## [Author Response · Author response to Decision Letter 1]

26 Sep 2022

A rebuttal letter has been attached

---

## [Decision Letter · Decision Letter 2]

1 Dec 2022

PONE-D-21-18907R2Access to HIV prevention, treatment, and care services during COVID-19 by men who have sex with men in Zimbabwe, An interpretive phenomenological analysis studyPLOS ONE

Dear Dr. Moyo,

Thank you for submitting your manuscript to PLOS ONE. After careful consideration, we feel that it has merit but does not fully meet PLOS ONE’s publication criteria as it currently stands. Therefore, we invite you to submit a revised version of the manuscript that addresses the points raised during the review process.

ACADEMIC EDITOR:Thank you for revising your paper. There are few additional comments for you to address. Please also pay attention to the English language which still requires further review and correction throughout.

We look forward to receiving your revised manuscript.

Kind regards,

Tanya Doherty, PhD

Academic Editor

PLOS ONE

Journal Requirements:

Reviewers' comments:

Reviewer's Responses to Questions

**Comments to the Author**

1. If the authors have adequately addressed your comments raised in a previous round of review and you feel that this manuscript is now acceptable for publication, you may indicate that here to bypass the “Comments to the Author” section, enter your conflict of interest statement in the “Confidential to Editor” section, and submit your "Accept" recommendation.

Reviewer #3: (No Response)

2. Is the manuscript technically sound, and do the data support the conclusions?

Reviewer #3: Yes

3. Has the statistical analysis been performed appropriately and rigorously? 

Reviewer #3: N/A

4. Have the authors made all data underlying the findings in their manuscript fully available?

Reviewer #3: No

5. Is the manuscript presented in an intelligible fashion and written in standard English?

Reviewer #3: Yes

6. Review Comments to the Author

Reviewer #3: The authors have presented a good manuscript that addresses most of the comments from the previous review. There are a few outstandingissues that need attention, which I am sure the authors will be able to speedily address.

1. Avoid phrases like "the MSM" in the abstract - this can be objectifiying by using "the".

2. Some grammatical issues are still in the manuscript e.g lines 149-150; 167-168; 186-187; 239; Table 1 (note: I am using line numbers in the Tracked Changed version of the manuscript).

3. Phrases like "hard-to-reach population" are still used in the manuscript (e.g. line 190), despite criticisms in the previous review and depite the authors claiming that "The term “ 'hard to reach' was replaced by the term “difficult to access”.

4.

On line 290 you note that the participants "self-identified as MSM" (presumably as per the inclusion criteria), however, further to this, did they also self-identify as "homosexual" or would they have preferred to be described as "gay"?

I ask this because there there are still references to "the gay community" (e.g. line 171). You need to add a footnote or in-text clarification about whether your participants self-identify as "gay" and/or "homosexual" and/or "MSM" (and/or how these categories of identity overlap, if at all). This is important, in order to be ethical and respectful of participants' identities. This speaks to the politics of representation in qualitative research.

7. PLOS authors have the option to publish the peer review history of their article (what does this mean?). If published, this will include your full peer review and any attached files.

Reviewer #3: **Yes: **Suntosh R. Pillay

---

## [Author Response · Author response to Decision Letter 2]

15 Jan 2023

Attached/uploaded copy of reviewer's comments

---

## [Editor Report · Decision Letter 3]

2 Feb 2023

Access to HIV prevention, treatment, and care services during COVID-19 by men who have sex with men in Zimbabwe, An interpretive phenomenological analysis study

PONE-D-21-18907R3

Dear Dr. Moyo,

We’re pleased to inform you that your manuscript has been judged scientifically suitable for publication and will be formally accepted for publication once it meets all outstanding technical requirements.

Kind regards,

Tanya Doherty, PhD

Academic Editor

PLOS ONE
---

## [Editor Report · Acceptance letter]

24 Feb 2023

PONE-D-21-18907R3 

Access to HIV prevention, treatment, and care services during COVID-19 by men who have sex with men in Zimbabwe, An interpretive phenomenological analysis study 

Dear Dr. Moyo:

I'm pleased to inform you that your manuscript has been deemed suitable for publication in PLOS ONE. Congratulations! Your manuscript is now with our production department. 

Kind regards, 

on behalf of

Professor Tanya Doherty 

Academic Editor

PLOS ONE